# Spatial Sign based Direct Sparse Linear Discriminant Analysis for High Dimensional Data

## Abstract

Robust high-dimensional classification under heavy-tailed distributions without losing efficiency, is a central challenge in modern statistics and machine learning. However, most existing linear discriminant analysis (LDA) methods are sensitive to deviations from normality and may suffer from suboptimal performance in heavy-tailed settings. This paper investigates the robust LDA problem with elliptical distributions in high-dimensional data. Our approach constructs stable discriminant directions by leveraging a robust spatial sign-based mean and covariance estimator, which allows accurate estimation even under extreme distributions. We demonstrate that SSLDA achieves an optimal convergence rate in terms of both misclassification rate and estimate error. Our theoretical results are further confirmed by extensive numerical experiments on both simulated and real datasets. Compared with state-of-the-art approaches, the SSLDA method offers superior improved finite sample performance and notable robustness against heavy-tailed distributions.

## 1 Introduction

High-dimensional data is increasingly prevalent in various real-world applications, such as genomic data(Schäfer & Strimmer, 2005), investment portfolio data(Ledoit & Wolf, 2003), and fMRl decoding(Shi et al., 2009). High-dimensional classification problems have garnered significant interest in recent decades. LDA is extensively utilized among numerous classification methods owing to its efficacy in practical applications. However, in high-dimensional settings, the classical LDA reveal the following critical limitations. Firstly, the precision matrix is either non-estimable or exceedingly challenging to estimate because to the singularity and irreversibility of the covariance matrix; Secondly, there is a compounding of errors in the estimation of unknown parameters. Bickel & Levina (2004) discovered that the efficacy of classical LDA may resemble that of random guessing in high-dimensional samples, despite the validity of the Gaussian assumption.

These limitations have resulted in the advancement of enhanced LDA techniques founded on certain sparse assumptions to address high-dimensional classification contexts. One typical strategy involves incorporating regularisation into the classification direction vector, such as the $\ell_1$ regularization (Witten & Tibshirani, 2009; Mai et al., 2012; Clemmensen et al., 2011) or the $\ell_2$ regularization (Guo et al., 2007), among others. Another specific strategy involves assuming that the covariance matrix $\Sigma$ and the mean difference $\delta = \mu_1 - \mu_2$ are sparse, hence facilitating consistent estimation of them. In Bickel & Levina (2004), a naive Bayes rule or the independence principle is presented by substituting $\Sigma$ with the diagonal of the sample covariance matrix. Shao et al. (2011) proposed a sparse LDA method based on thresholding methodology and demonstrated the resultant can theoretically achieve the Bayes error. Tibshirani et al. (2002) and Fan & Fan (2008) proposed the nearest shrunken centroid estimation and the features annealed independent rule, wherein variable selection is executed through soft and hard thresholding criteria, respectively.

In contrast to the aforementioned approaches that separately estimation of $\Sigma^{-1}$ and $\delta$, some straightforward and efficient classifiers are introduced that directly estimate the product $\beta^* = \Sigma^{-1}\delta$ by assumes the sparsity of the discriminant direction. Cai & Liu (2011) introduced the linear programming discriminant rule for sparse linear discriminant analysis based on direct estimation of $\beta^*$ via limited $\ell_1$ minimization. The direct sparse discriminant analysis investigated in Mai et al. (2012) represents another widely utilised sparse LDA approach, which is both computationally efficient and relatively straightforward to comprehend, as it reformulates the high-dimensional LDA into a

penalised linear regression framework. To address the lack of adaptivity and theoretical guarantees in existing high-dimensional LDA methods, Cai & Zhang (2019) proposed an adaptive and tuning-free procedure that accounts for heteroscedasticity and achieves minimax optimality. In general, direct estimate $\beta^*$ offers a considerable computational advantage over existing approaches that necessitate separate estimations of them; it is more economical and has strong performance, even when covariance or mean differences are not consistently assessed.

Note that LDA shows high sensitivity to outliers (Hastie, 2009), particularly in high-dimensional datasets. Robust estimation of the mean and covariance is essential for the efficacy of LDA. Optimized LDA performance occurs when normality and homoscedasticity are met (Croux et al., 2008). It is critical to point out that traditional robust mean estimation techniques, such as the coordinate-wise median and geometric median, experience degradation in high-dimensional spaces, as their error bounds are proportional to the dimensionality. Numerous research investigate robust mean estimate for tainted data in high dimensions, such as Tukey's median and Iterative Filtering(Diakonikolas et al., 2019; 2017). Recent works have concentrated on reducing the spectral norm of weighted sample covariance and estimating the mean by a weighted average (Cheng et al., 2019; Zhu et al., 2022). Motivated by this methodology, in Deshmukh et al. (2023), the $\ell_1$ norm of an outlier indicator vector is reduced subject to a constraint on the spectral norm of the weighted sample covariance, resulting in an order-optimal robust estimate of the mean. Based on $\ell_1$ norm, Li et al. (2019) introduces a robust discriminant analysis criterion, which is an upper bound of the theoretical framework of Bhattacharyya optimality.

However, most existing methods work well under (sub-) Gaussian assumptions but poorly on heavy-tailed distributions. For more general distributions, Fang & Anderson (1990) demonstrated that the Fisher rule remains best for elliptical distributions, which encompasses multivariate normal-$t$ and double exponential distributions. Wakaki (1994) explored Fisher's linear discriminant function for a wide class of elliptical symmetric distributions sharing a common covariance matrix. To improve discriminant analysis method robustness and efficiency, Andrews et al. (2011) examined linear and quadratic discriminant classification with a mixture of multivariate-$t$ distributions. Bose et al. (2015) endeavored to extend linear and quadratic discriminant analysis to elliptically symmetric distributions. Some studies have investigated the efficacy of classification and theoretical intricacies associated with elliptical distributions (e.g., Cai & Liu (2011); Shao et al. (2011); Yang et al. (2023)). Han et al. Han et al. (2013) successfully generalized these principles to broader distributions utilizing the Gaussian linkage approach. Recently, by allowing each observation to originate from its own elliptically symmetric distribution, Houdouin et al. (2024) develops an innovative, robust, and model-free discriminant analysis algorithm.

Within the framework of elliptical distributions, employing a spatial-sign-based methodology has shown significant effectiveness, even in high-dimensional situations (Oja, 2010; Raninen et al., 2021). Under high-dimensional elliptical populations, Li & Zhou (2017) investigates spatial-sign covariance matrix on its asymptotic spectral behaviors. Wang et al. (2015) introduced a nonparametric one-sample test utilizing the multivariate spatial sign transformation for elliptically distributed data. Based on spatial ranks and inner standardization, Feng & Sun (2016) present spatial-sign-based test methodologies for high-dimensional one-sample localization problems. In the context of the two-sample location problem, Feng et al. (2016) proposed a robust multivariate-sign-based procedures and Huang et al. (2023) advocated for a two-sample inverse norm sign test. See Zou et al. (2014), Feng & Liu (2017), Zhang et al. (2022), Feng (2024), and the references therein for more related studies on spatial sign-based approaches.

In this work, we introduce a high-dimensional sparse LDA method designed to directly estimate the 'discriminant direction' under the assumption of elliptical distribution. The main contributions of the paper are summarized as below.

- We establish theoretical results for Spatial-sign based linear discriminant analysis (SSLDA) in the sparse scenario.

- We show the consistency and rate of convergence results both misclassification rate and estimate error by assuming that the data is elliptically distributed.

- We employ the sample spatial median and the spatial sign covariance matrix by directly estimate 'discriminant direction', which we demonstrate to be robust and efficient under fairly general assumptions.

- Empirical studies are employed to evaluate image classification performance using datasets of braeburn apples and white cabbages. Results demonstrate that the proposed approach outperforms existing methods, showing promising performance in accuracy and efficiency.
- Code is available at https://github.com/RobustC/SSLDA, supporting real and complex-valued data.

The rest of the paper is organized as follows. A robust classification method, named SSLDA, is present in Section 2. In Section 3, we investigate the asymptotic properties of SSLDA. Section 4 showcases the results of numerical simulations, while Section 5 demonstrates a real-world application to image classification. Finally, discussion and limitation are present in Section 6.

*Notation:* For a vector $\boldsymbol{v} = (v_1, \ldots, v_p)'$, we define $\|\boldsymbol{v}\|_0 = \sum_{j=1}^p \mathbb{1}\{v_j \neq 0\}$, $\|\boldsymbol{v}\|_1 = \sum_{j=1}^p |v_j|$ and $\|\boldsymbol{v}\|_2 = \sqrt{\sum_{j=1}^p v_j^2}$, $\|\boldsymbol{v}\|_\infty = \max_{1 \leq j \leq p} |v_j|$, where $\mathbb{1}(\cdot)$ is the indicator function that returns 1 if the condition inside the brackets is true, and 0 otherwise. For a matrix $\mathbf{M} = (a_{ij})_{p \times q}$, we define the matrix $\ell_1$ norm $\|\mathbf{M}\|_{L_1} = \max_{1 \leq j \leq q} \sum_{i=1}^p |a_{ij}|$, the elementwise $\ell_\infty$ norm $\|\mathbf{M}\|_\infty = \max\{|a_{ij}|\}$. $\lambda_{\min}(\mathbf{M})$ and $\lambda_{\max}(\mathbf{M})$ denote the smallest and largest eigenvalues of $\mathbf{M}$. $a_n \asymp b_n$ signifies $a_n = O(b_n)$ and $b_n = O(a_n)$ for any positive number sequences $\{a_n\}$ and $\{b_n\}$. For any random variable $X \in \mathbb{R}$, we define the sub-Gaussian norm as $\|X\|_{\psi_2} := \sup_{k \geq 1} k^{-1/2} \left(E|X|^k\right)^{1/k}$.

## 2 ROBUST CLASSIFICATION METHOD

In the context of classifying between two $p$-dimensional normal distributions, $N(\boldsymbol{\mu}_1, \boldsymbol{\Sigma})$ (designated as class 1) and $N(\boldsymbol{\mu}_2, \boldsymbol{\Sigma})$ (designated as class 2), both sharing the same covariance matrix $\boldsymbol{\Sigma}$, we consider a random vector $\mathbf{Z}$ that originates from one of these distributions with equal prior probabilities. The task is to determine the class to which $\mathbf{Z}$ belongs. When the parameters $\boldsymbol{\mu}_1$, $\boldsymbol{\mu}_2$, and $\boldsymbol{\Sigma}$ are known, Fisher's linear discriminant rule provides a straightforward solution. This rule is given by:

$$\psi_{\mathbf{F}}(\mathbf{Z}) = \mathbb{1}\left\{(\mathbf{Z} - \boldsymbol{\mu})'\boldsymbol{\Omega}\boldsymbol{\delta} \geq 0\right\},$$

where $\boldsymbol{\mu} = \frac{\boldsymbol{\mu}_1 + \boldsymbol{\mu}_2}{2}$ is the midpoint between the two mean vectors, $\boldsymbol{\delta} = \boldsymbol{\mu}_1 - \boldsymbol{\mu}_2$ is the difference between the two mean vectors, $\boldsymbol{\Omega} = \boldsymbol{\Sigma}^{-1}$ is the inverse of the covariance matrix.

According to this rule, $\mathbf{Z}$ is classified into class 1 if $\psi_{\mathbf{F}}(\mathbf{Z}) = 1$, and into class 2 otherwise. Fisher's linear discriminant is the optimal classifier in this scenario, as it coincides with the Bayes rule when the prior probabilities for the two classes are equal.

In practice, we often don't know the true parameters, so we estimate them using samples. Suppose $\{\mathbf{X}_k; 1 \leq k \leq n_1\}$ and $\{\mathbf{Y}_k; 1 \leq k \leq n_2\}$ are independent and identically distributed random samples from $N(\boldsymbol{\mu}_1, \boldsymbol{\Sigma})$ and $N(\boldsymbol{\mu}_2, \boldsymbol{\Sigma})$, respectively. Set $\hat{\boldsymbol{\mu}}_1$ and $\hat{\boldsymbol{\mu}}_2$ are the sample means of these two samples, respectively. And $\hat{\boldsymbol{\Sigma}} = \frac{1}{n}\left(n_1\hat{\boldsymbol{\Sigma}}_1 + n_2\hat{\boldsymbol{\Sigma}}_2\right)$, $n = n_1 + n_2$ where $\hat{\boldsymbol{\Sigma}}_1, \hat{\boldsymbol{\Sigma}}_2$ are the sample covariance matrix of these two samples, respectively. So $\mathbf{Z}$ is classified into class 1 if

$$(\mathbf{Z} - \hat{\boldsymbol{\mu}})'\hat{\boldsymbol{\Omega}}\hat{\boldsymbol{\delta}} \geq 0$$

where $\hat{\boldsymbol{\mu}} = \frac{\hat{\boldsymbol{\mu}}_1 + \hat{\boldsymbol{\mu}}_2}{2}$, $\hat{\boldsymbol{\delta}} = \hat{\boldsymbol{\mu}}_1 - \hat{\boldsymbol{\mu}}_2$ and $\hat{\boldsymbol{\Omega}} = \hat{\boldsymbol{\Sigma}}^{-1}$. When the dimension $p$ is larger than the sample sizes, $\hat{\boldsymbol{\Sigma}}$ is not invertible. So the above classification rule can not work well. Consequently, many literatures consider the high dimensional linear discriminant analysis, such as Cai & Liu (2011); Le et al. (2020); Park et al. (2022). In an important work, Cai & Liu (2011) proposed a direct estimation approach to sparse linear discriminant analysis. They estimate $\boldsymbol{\beta}^* = \boldsymbol{\Omega}\boldsymbol{\delta}$ directly by the solution to the following optimization problem:

$$\hat{\boldsymbol{\beta}} \in \underset{\boldsymbol{\beta} \in \mathbb{R}^p}{\arg\min}\left\{\|\boldsymbol{\beta}\|_1 \text{ subject to } \left\|\hat{\boldsymbol{\Sigma}}\boldsymbol{\beta} - (\hat{\boldsymbol{\mu}}_1 - \hat{\boldsymbol{\mu}}_2)\right\|_\infty \leq \lambda_n\right\}, \tag{1}$$

where $\lambda_n$ is a tuning parameter. The constrained $\ell_1$ minimization method (1) is known to be an effective way for reconstructing sparse signals, see Donoho et al. (2005) and Candes & Tao (2007). Then, they proposed a new classification rule: $\mathbf{Z}$ is classified into class 1 if

$$(\mathbf{Z} - \hat{\boldsymbol{\mu}})'\hat{\boldsymbol{\beta}} \geq 0.$$

However, the above methods are all constructed based on the sample means and covariance matrix which do not perform very well for heavy-tailed distributions. In this paper, we assume $\boldsymbol{X}$ and $\boldsymbol{Y}$ are generated from the elliptical distribution with density

$$|\boldsymbol{\Lambda}|^{-1/2} g\left((x - \boldsymbol{\mu}_k)'\boldsymbol{\Lambda}^{-1}(x - \boldsymbol{\mu}_k)\right), k = 1, 2,$$

respectively, where $g(\cdot)$ is a decreasing function. Without loss of generality, we assume that $\mathrm{tr}(\boldsymbol{\Lambda}) = p$. If the covariance matrix $\boldsymbol{\Sigma} = \mathrm{Cov}(\boldsymbol{X}) = \mathrm{Cov}(\boldsymbol{Y})$ exist, $\boldsymbol{\Sigma} = \omega\boldsymbol{\Lambda}$ with positive parameter $\omega = p^{-1}\mathrm{tr}(\boldsymbol{\Sigma}) \in \mathbb{R}$.Fang & Anderson (1990) showed that Fisher's rule is still optimal for elliptical distributions. Since the constant $\omega$ does not affect the decision rule, we can, without loss of generality, set $\omega = 1$. In this case, we use the sample spatial median and spatial-sign covariance matrix to replace the sample mean and covariance matrix.

We often use the spatial median to estimate $\boldsymbol{\mu}$, i.e.

$$\tilde{\boldsymbol{\mu}}_1 = \arg\min_{\boldsymbol{\mu} \in \mathbb{R}^p} \sum_{i=1}^{n_1} \|\boldsymbol{X}_i - \boldsymbol{\mu}\|_2, \; \tilde{\boldsymbol{\mu}}_2 = \arg\min_{\boldsymbol{\mu} \in \mathbb{R}^p} \sum_{i=1}^{n_2} \|\boldsymbol{Y}_i - \boldsymbol{\mu}\|_2 \tag{2}$$

Let $U(\boldsymbol{x}) = \frac{\boldsymbol{x}}{\|\boldsymbol{x}\|_2}I(\boldsymbol{x} \neq \boldsymbol{0})$. Then the sample spatial sign covariance matrix is defined as

$$\hat{\mathbf{S}}_1 = \frac{1}{n_1}\sum_{i=1}^{n_1} U(\boldsymbol{X}_i - \tilde{\boldsymbol{\mu}}_1)U(\boldsymbol{X}_i - \tilde{\boldsymbol{\mu}}_1)', \; \hat{\mathbf{S}}_2 = \frac{1}{n_2}\sum_{i=1}^{n_2} U(\boldsymbol{Y}_i - \tilde{\boldsymbol{\mu}}_2)U(\boldsymbol{Y}_i - \tilde{\boldsymbol{\mu}}_2)' \tag{3}$$

and the population spatial-sign covariance matrix is estimated by $\hat{\mathbf{S}} = \frac{1}{n}\left(n_1\hat{\mathbf{S}}_1 + n_2\hat{\mathbf{S}}_2\right)$. We estimate $\boldsymbol{\gamma}^* = \boldsymbol{\Lambda}^{-1}\boldsymbol{\delta}$ directly by the solution to the following optimization problem:

$$\hat{\boldsymbol{\gamma}} \in \arg\min_{\boldsymbol{\gamma} \in \mathbb{R}^p}\left\{\|\boldsymbol{\gamma}\|_1 \text{ subject to } \left\|p\hat{\mathbf{S}}\boldsymbol{\gamma} - (\tilde{\boldsymbol{\mu}}_1 - \tilde{\boldsymbol{\mu}}_2)\right\|_\infty \leq \lambda_n\right\}, \tag{4}$$

and the corresponding classification rule is $\mathbf{Z}$ is classified into class 1 if

$$(\mathbf{Z} - \tilde{\boldsymbol{\mu}})'\hat{\boldsymbol{\gamma}} \geq 0. \tag{5}$$

where $\tilde{\boldsymbol{\mu}} = \frac{\tilde{\boldsymbol{\mu}}_1 + \tilde{\boldsymbol{\mu}}_2}{2}$.

## 3 THEORETICAL RESULTS

The optimal misclassification rate in this case is

$$R := \frac{1}{2}\mathrm{P}\left((\boldsymbol{X} - \boldsymbol{\mu}_1)'\boldsymbol{\Omega}\boldsymbol{\delta} < -\frac{1}{2}\boldsymbol{\delta}'\boldsymbol{\Omega}\boldsymbol{\delta}\right) + \frac{1}{2}\mathrm{P}\left((\boldsymbol{Y} - \boldsymbol{\mu}_2)'\boldsymbol{\Omega}\boldsymbol{\delta} \geq \frac{1}{2}\boldsymbol{\delta}'\boldsymbol{\Omega}\boldsymbol{\delta}\right).$$

As in the work of Shao et al. (2011), under the assumption of elliptical distribution, for any $p$-dimensional non-random vector $\boldsymbol{u}$ with $\|\boldsymbol{u}\|_2 = 1$ and any $t \in \mathbb{R}$,

$$\mathrm{P}\left(\boldsymbol{u}'\boldsymbol{\Omega}^{1/2}(\boldsymbol{X} - \boldsymbol{\mu}_1) \leq t\right) =: \Psi(t)$$

is a continuous distribution function symmetric about 0 and does not depend on $\boldsymbol{u}$. Given $\{\mathbf{X}_k\}$ and $\{\mathbf{Y}_k\}$, the conditional classification error of the linear programming discriminant (LPD) rule is

$$R_n := 1 - \frac{1}{2}\Psi\left(-\frac{(\tilde{\boldsymbol{\mu}} - \boldsymbol{\mu}_1)'\hat{\boldsymbol{\gamma}}}{(\hat{\boldsymbol{\gamma}}'\boldsymbol{\Sigma}\hat{\boldsymbol{\gamma}})^{1/2}}\right) - \frac{1}{2}\Psi\left(\frac{(\tilde{\boldsymbol{\mu}} - \boldsymbol{\mu}_2)'\hat{\boldsymbol{\gamma}}}{(\hat{\boldsymbol{\gamma}}'\boldsymbol{\Sigma}\hat{\boldsymbol{\gamma}})^{1/2}}\right).$$

where $\hat{\boldsymbol{\gamma}}$ is given in (4). The efficacy of the LPD rule can be effectively gauged through the difference (or ratio) between $R_n$ and $R$. Let $\Delta_p = \boldsymbol{\delta}'\boldsymbol{\Omega}\boldsymbol{\delta} = \omega^{-1}\boldsymbol{\delta}'\boldsymbol{\Lambda}^{-1}\boldsymbol{\delta}$. $\{\sigma_{ii} = \boldsymbol{\Sigma}_{ii}\}_{i=1}^p$ denote the corresponding marginal variances. Refer to Cai & Liu (2011), we need provide the following conditions before presenting the difference and ratio between $R_n$ and $R$.

(C1) $n_1 \asymp n_2, \log p \leq n, c_0^{-1} \leq \lambda_{\min}(\boldsymbol{\Sigma}) \leq \lambda_{\max}(\boldsymbol{\Sigma}) \leq c_0, \max_{1 \leq i \leq p} \sigma_{ii} \leq K$ and $\Delta_p \geq c_1$ for some constant $K > 0$ and $c_0, c_1 > 0$.

(C2) Define $\zeta_k = \mathbb{E}\left(\xi_i^{-k}\right), \xi_i = \|\boldsymbol{X}_i - \boldsymbol{\mu}\|_2, \nu_i = \zeta_1^{-1}\xi_i^{-1}$. (1) $\zeta_k\zeta_1^{-k} < \zeta \in (0, \infty)$ for $k = 1, 2, 3, 4$ and all $p$. (2) $\limsup_p \lambda_{\max}(\mathbf{S}) < 1 - \psi < 1$ for some positive constant $\psi$. (3) $\nu_i$ is sub-gaussian distributed, i.e. $\|\nu_i\|_{\psi_2} \leq K_\nu < \infty$.

Condition (C1) is the same as condition (C1) in Cai & Liu (2011), which is commonly used conditions in the high dimensional setting. Condition (C2) are consistent with conditions (A1-A2) in Feng (2024), which ensure the consistency of the spatial median estimator (2).

Then, we having the following theoretical results.

**THEOREM 3.1.** *Let $\lambda_n = C\sqrt{\Delta_p \log p / n}$ with $C$ being a sufficiently large constant. Suppose (C1)-(C2) hold and*

$$\frac{\|\boldsymbol{\Lambda}^{-1}\boldsymbol{\delta}\|_1}{\Delta_p^{1/2}} + \frac{\|\boldsymbol{\Lambda}^{-1}\boldsymbol{\delta}\|_1^2}{\Delta_p^2} = o\left(\sqrt{\frac{n}{\log p}}\right). \tag{6}$$

*Then we have as $n \to \infty$ and $p \to \infty$,*

$$R_n - R \to 0$$

*in probability.*

**THEOREM 3.2.** *Let $\lambda_n = C\sqrt{\Delta_p \log p / n}$ with $C$ being a sufficiently large constant and $\frac{n}{p \log p} \to 0$. Suppose (C1)-(C2) hold and*

$$\|\boldsymbol{\Lambda}^{-1}\boldsymbol{\delta}\|_1\Delta_p^{1/2} + \|\boldsymbol{\Lambda}^{-1}\boldsymbol{\delta}\|_1^2 = o\left(\sqrt{\frac{n}{\log p}}\right).$$

*Then*

$$\frac{R_n}{R} - 1 = O\left(\left(\|\boldsymbol{\Lambda}^{-1}\boldsymbol{\delta}\|_1\Delta_p^{1/2} + \|\boldsymbol{\Lambda}^{-1}\boldsymbol{\delta}\|_1^2\right)\sqrt{\frac{\log p}{n}}\right)$$

*with probability greater than $1 - O\left(p^{-1}\right)$. In particular, if (C1)-(C2) hold and*

$$\|\boldsymbol{\Lambda}^{-1}\boldsymbol{\delta}\|_0\Delta_p = o\left(\sqrt{\frac{n}{\log p}}\right),$$

*then*

$$\frac{R_n}{R} - 1 = O\left(\|\boldsymbol{\Lambda}^{-1}\delta\|_0\Delta_p\sqrt{\frac{\log p}{n}}\right)$$

*with probability greater than $1 - O\left(p^{-1}\right)$.*

Theorem 3.1 and 3.2 is similar to the results in Theorem 2 and 4 in Cai & Liu (2011). Theorem 3.1 show the consistency of our proposed method and Theorem 3.2 establish the rate of convergence.

## 4 SIMULATION STUDIES

In this section, we investigate the empirical performance of the SSLDA method.

### 4.1 IMPLEMENTATION OF SSLDA

The estimate of $\boldsymbol{\gamma}^*$ is obtained by solving $\ell_1$ minimization problem of (4). This convex optimazation problem can be reformulated as the following linear program

$$\min \sum_{j=1}^{p} u_j$$

$$\begin{aligned}
\text{subject to:} &- \gamma_j \leq u_j \quad \text{for all} 1 \leq j \leq p, \\
&+ \gamma_j \leq u_j \quad \text{for all} 1 \leq j \leq p, \\
&- p\hat{\boldsymbol{\sigma}}_k'\hat{\boldsymbol{\gamma}}_k + \tilde{\boldsymbol{\delta}}_k \leq \lambda_n \quad \text{for all} 1 \leq k \leq p, \\
&+ p\hat{\boldsymbol{\sigma}}_k'\hat{\boldsymbol{\gamma}}_k + \tilde{\boldsymbol{\delta}}_k \leq \lambda_n \quad \text{for all} 1 \leq k \leq p,
\end{aligned} \tag{7}$$

where $(\tilde{\delta}_1, \cdots, \tilde{\delta}_p) := \tilde{\boldsymbol{\delta}}$ and $(\hat{\boldsymbol{\sigma}}_1, \cdots, \hat{\boldsymbol{\sigma}}_p) := \hat{\mathbf{S}}$. We applied the CLIME method (Cai et al., 2011) to solve (7). We replace $\hat{\mathbf{S}}$ in (4) by $\hat{\mathbf{S}}_\rho = \hat{\mathbf{S}} + \rho \boldsymbol{I}_{p \times p}$ with a small positive number $\rho$ (e.g., $\rho = \sqrt{\log p / n}$).

The algorithm's tuning parameter $\lambda = \lambda_n$ can be optimized through empirical 10-fold cross-validation (CV). To implement this, partition the sets $\{1, 2, \cdots, n_1\}$ and $\{1, 2, \cdots, n_2\}$ into $2K$ subgroups $G_{ik}$, where $i = 1, 2, k = 1, 2, \cdots, K$. This division naturally separates the sample data $\{\mathbf{X}_i; 1 \leq i \leq n_1\}$ and $\{\mathbf{Y}_j; 1 \leq j \leq n_2\}$ into $K$ validation subsets $\mathscr{X}_k := \{\mathbf{X}_i, \mathbf{Y}_j : i \in G_{1k}, j \in G_{2k}\}, 1 \leq k \leq K$. Denote $\tilde{\boldsymbol{\mu}}_{(k)}, \hat{\boldsymbol{S}}_{(k)}$ be defined in (2) and (3) derived from $\{\mathbf{X}_k, \mathbf{Y}_k; 1 \leq k \leq n_1\} \backslash \mathscr{X}_k$. Based on $\tilde{\boldsymbol{\mu}}_{(k)}$, $\hat{\boldsymbol{S}}_{(k)}$, we can obtain $\hat{\boldsymbol{\gamma}}$ for a given $\lambda_n$ by (4). The final selection of $\lambda$ boils down to

$$\hat{\lambda} = \max_\lambda \sum_{k=1}^{K} \left( \sum_{i \in G_{1k}} I_{i1}^{(k)} + \sum_{j \in G_{2k}} I_{j2}^{(k)} \right),$$

where $I_{j1}^{(k)} = 1$ if $\mathbf{X}_i \in \mathscr{X}_k$ stisfies (5), else $I_{j1}^{(k)} = 0$; let $I_{j2}^{(k)} = 1$ if $\mathbf{Y}_j \in \mathscr{X}_k$ not stisfies (5), else $I_{j2}^{(k)} = 0$, and we choose $K = 10$ in this paper.

## 4.2 Simulation Results

We compare the numerical performance of the SSLDA method with fellowing methods:

- LS-LDA: The least square formulation for classification proposed by Mai et al. (2012).
- CODA: Copula discriminant analysis classifier (CODA) for high-dimensional data proposed by Han et al. (2013).
- LDA-CLIME: Linear programming discriminant for high-dimensional data clssification using CLIME method proposed by Cai et al. (2011).
- AdaLDA: An adaptive algorithm (Cai & Zhang, 2019) for high dimensional LDA.
- SSLDA: Sparse spatial-sign based linear discriminant analysis.

In the simulation studies, we fix the sample sizes $n = 400$ and varied $p$ to be $\{100, 200, 300, 500, 800\}$. Let $\boldsymbol{\mu}_1 = \boldsymbol{0}, \boldsymbol{\mu}_2 = (1, \cdots, 1, 0, \cdots, 0)$, where the number of 1's is $s_0 = 10$. For each class $g$ $(g = 1, 2)$, the $p$-dimensional predictors $x$ are sampled from the following four elliptical distributions:

(I) Multivariate normal distribution: $\mathbf{Z} \sim N(\boldsymbol{\mu}_g, \boldsymbol{\Sigma})$.

(II) Multivariate $t_2$-distribution, data are generated from standardized $\frac{t_2}{\sqrt{2}}$ with mean $\boldsymbol{\mu}_g$ and $\boldsymbol{\Sigma}$.

(III) Standardized multivariate mixture normal distribution $MN_{N,\kappa,10} = [\kappa N(\boldsymbol{\mu}_g, \boldsymbol{\Sigma}) + (1 - \kappa)N(\boldsymbol{\mu}_g, 10^2 \boldsymbol{\Sigma})] / \sqrt{\kappa + 10^2(1 - \kappa)}$. $\kappa$ is chosen to be 0.8.

(IV) Cauchy distribution.

Based on the above four distributions, we considered the following two models.

- **Model 1**: $\boldsymbol{\Sigma}_{i,j} = 1 - 0.5 \times \mathbb{1}\{|i - j| \neq 0\}, i, j = 1, \cdots, p$.
- **Model 2**: $\boldsymbol{\Sigma}_{i,j} = 0.8^{|i-j|}, i, j = 1, \cdots, p$.

In line with the simulation parameters outlined in Cai & Liu (2011), this study fixes $n_1 = n_2 = 200$. The average classification errors for the test samples and the standard deviations based on 100 replications are reported in Table 1 and Table 2. These tables present the performance of SSLDA in comparison with four state-of-the-art LDA variants and similarity-based methods. In addition, we also compared SSLDA with several widely used general-purpose machine learning classifiers, and the simulation results for these classifiers are provided in the Appendix C. Table 1 and Table 2 summarize results across four distinct distributions of varying dimensionality, which serve to characterize the level of noise in the original data distribution. The results show that SSLDA consistently outperforms both the LDA-based baselines and the general-purpose classifiers under both Model 1 and Model

Table 1: The average classification error and the standard error (in brackets) for the test samples in percentage for Model 1 over 100 Monte Carlo replications. (%)

| Distribution | $p$ | SSLDA | LDA-CLIME | CODA | LS-LDA | AdaLDA |
|---|---|---|---|---|---|---|
| Normal | 100 | **2.46(0.087)** | 2.93(0.088) | 2.75(0.083) | 2.98(0.093) | 3.70(0.103) |
| | 200 | **2.26(0.072)** | 2.60(0.076) | 2.68(0.086) | 2.71(0.082) | 3.55(0.089) |
| | 300 | **2.26(0.073)** | 2.31(0.074) | 2.34(0.077) | 2.57(0.083) | 3.42(0.095) |
| | 500 | 2.19(0.078) | **2.17(0.090)** | 2.37(0.092) | 2.34(0.084) | 3.30(0.093) |
| | 800 | **2.14(0.082)** | 2.65(0.085) | 2.48(0.094) | 2.20(0.076) | 3.11(0.093) |
| $t_2$ | 100 | **9.36(0.151)** | 10.91(0.156) | 10.84(0.149) | 11.26(0.190) | 18.01(0.669) |
| | 200 | **9.23(0.141)** | 10.81(0.160) | 10.40(0.172) | 10.62(0.166) | 18.30(0.539) |
| | 300 | **8.95(0.122)** | 10.65(0.160) | 10.64(0.191) | 10.75(0.194) | 18.80(0.592) |
| | 500 | **8.89(0.171)** | 11.05(0.155) | 10.26(0.170) | 9.77(0.156) | 18.57(0.586) |
| | 800 | **9.24(0.118)** | 12.55(0.184) | 10.11(0.123) | 9.95(0.172) | 18.06(0.573) |
| $MN_{N,\kappa,10}$ | 100 | **10.78(0.167)** | 15.17(0.228) | 12.28(0.174) | 16.69(0.246) | 25.27(0.489) |
| | 200 | **10.14(0.128)** | 13.91(0.211) | 11.95(0.166) | 13.66(0.187) | 26.85(0.438) |
| | 300 | **10.47(0.149)** | 12.64(0.163) | 11.37(0.154) | 12.57(0.179) | 27.40(0.475) |
| | 500 | **10.37(0.148)** | 12.31(0.159) | 11.96(0.177) | 11.67(0.172) | 28.34(0.505) |
| | 800 | **10.22(0.186)** | 12.20(0.170) | 11.43(0.175) | 11.52(0.161) | 30.52(0.648) |
| Cauchy | 100 | **15.43(0.179)** | 19.43(0.251) | 25.81(1.271) | 20.60(0.843) | 48.96(0.410) |
| | 200 | **15.69(0.208)** | 19.19(0.203) | 24.28(1.047) | 18.19(0.252) | 49.51(0.342) |
| | 300 | **16.77(0.209)** | 20.11(0.183) | 25.45(1.215) | 17.53(0.240) | 49.93(0.220) |
| | 500 | **15.05(0.202)** | 18.45(0.162) | 23.69(1.101) | 18.59(0.511) | 50.01(0.347) |
| | 800 | **15.24(0.146)** | 19.00(0.177) | 25.37(1.187) | 17.94(0.502) | 48.75(0.268) |

Table 2: The average classification error and the standard error (in brackets) for the test samples in percentage for Model 2 over 100 Monte Carlo replications. (%)

| Distribution | $p$ | SSLDA | LDA-CLIME | CODA | LS-LDA | AdaLDA |
|---|---|---|---|---|---|---|
| Normal | 100 | **17.78(0.213)** | 18.00(0.186) | 18.08(0.214) | 17.93(0.197) | 19.21(0.273) |
| | 200 | **17.57(0.207)** | 20.11(0.218) | 18.01(0.216) | 18.51(0.227) | 19.33(0.219) |
| | 300 | 18.52(0.205) | 21.65(0.206) | 18.59(0.177) | **18.50(0.201)** | 19.09(0.237) |
| | 500 | 18.44(0.178) | 21.65(0.206) | 18.35(0.189) | **16.87(0.207)** | 19.32(0.214) |
| | 800 | 18.80(0.161) | 25.02(0.243) | 18.96(0.184) | **17.44(0.205)** | 19.62(0.223) |
| $t_2$ | 100 | **22.95(0.215)** | 26.82(0.280) | 24.60(0.229) | 26.28(0.281) | 30.89(0.529) |
| | 200 | **24.35(0.200)** | 27.46(0.243) | 25.73(0.263) | 26.82(0.296) | 31.94(0.448) |
| | 300 | **24.31(0.272)** | 28.68(0.245) | 25.79(0.296) | 27.12(0.325) | 31.98(0.356) |
| | 500 | **23.18(0.218)** | 30.85(0.267) | 26.43(0.286) | 26.31(0.261) | 31.47(0.381) |
| | 800 | **23.66(0.253)** | 32.32(0.252) | 27.30(0.256) | 27.18(0.248) | 34.71(0.636) |
| $MN_{N,\kappa,10}$ | 100 | **24.61(0.227)** | 33.57(0.382) | 26.62(0.266) | 31.23(0.258) | 34.98(0.524) |
| | 200 | **24.21(0.235)** | 35.86(0.393) | 27.45(0.281) | 29.28(0.259) | 37.66(0.560) |
| | 300 | **24.66(0.213)** | 36.10(0.438) | 28.18(0.338) | 28.64(0.259) | 39.32(0.604) |
| | 500 | **25.75(0.224)** | 32.74(0.330) | 29.19(0.305) | 28.66(0.273) | 39.96(0.578) |
| | 800 | **26.02(0.223)** | 31.30(0.279) | 29.80(0.357) | 28.54(0.265) | 39.64(0.544) |
| Cauchy | 100 | **26.75(0.230)** | 35.05(0.316) | 37.78(0.796) | 33.93(0.355) | 47.59(0.559) |
| | 200 | **26.74(0.233)** | 34.28(0.303) | 38.18(0.751) | 35.62(0.962) | 49.78(0.264) |
| | 300 | **26.77(0.186)** | 36.07(0.354) | 38.47(0.681) | 35.58(0.718) | 48.64(0.399) |
| | 500 | **27.35(0.250)** | 36.62(0.299) | 42.26(0.612) | 33.11(0.193) | 49.50(0.240) |
| | 800 | **26.48(0.239)** | 38.68(0.240) | 38.12(0.441) | 33.60(0.182) | 49.20(0.344) |

2. Moreover, as the tail of the original distribution thickens, the advantage of SSLDA becomes more pronounced, particularly for the Cauchy distribution, highlighting its robustness in challenging high-dimensional settings.

To further investigate the impact of truly influential features number on classification performance, we fixed $n_1 = n_2 = 200$ and $p = 100$, and examined the classification error rates of four distinct methods across different $s_0$ values. We run the four methods on each $s_0 \in [5, 80]$, each repeated for 100 times. The averaged misclassification errors in percentage versus various $s_0$ are illustrated in Figure 1. It can be observed in Figure 1 that SSLDA performs the best (blue curve) in different distributions, especially in heavy-tailed distributions (such as Cauchy distribution). The experimental results further highlight the superiority of the SSLDA method under high-dimensional heavy-tailed settings, demonstrating its versatility in handling both sparse and dense mean vector configurations with equal effectiveness.

In addition to classification accuracy, we also compared the computational efficiency of SSLDA with the competing methods. The running time results, presented in the Appendix C, show that SSLDA is efficient, and in some cases is even faster than certain competitors.

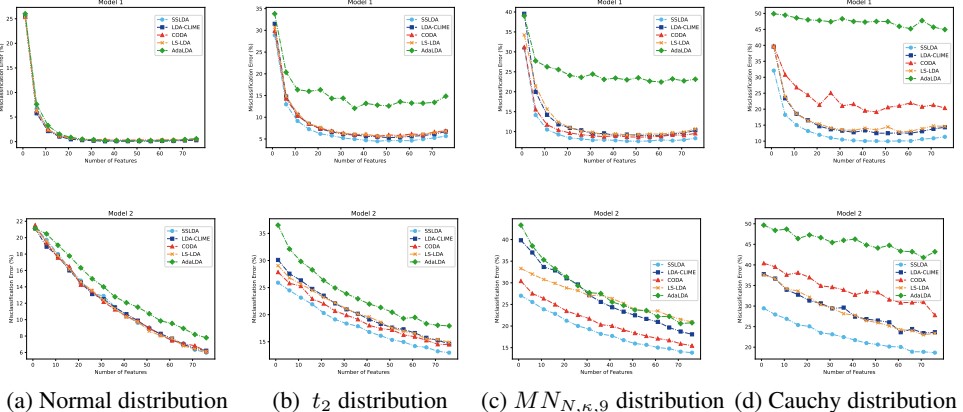

(a) Normal distribution     (b) $t_2$ distribution     (c) $MN_{N,\kappa,9}$ distribution     (d) Cauchy distribution

Figure 1: Average misclassification errors of five methods over 100 Monte Carlo replications ($n_1 = n_2 = 200, p = 100$) for different numbers of features $s_0$ and distributions. Top row: Model 1; Bottom row: Model 2.

## 5   REAL DATA APPLICATION

In this section, we apply the SSLDA classifier to the analysis of the real dataset to further examine the performance of the proposed rule.

**Datasets.** There are 636 JPG images of two groups, braeburn apple and white cabbage, with quantities of 492 and 144 (i.e. $n_1 = 492, n_2 = 144$), respectively. All images are 100×100 pixels in size. This image set is sourced from https://www.kaggle.com/datasets/moltean/fruits. The task of this dataset is to perform image classification on these images of Braeburn apples and white cabbages, to correctly distinguish between apples and cabbages. Figure 2 shows example images from each group along with the dataset's preprocessing stages. Each RGB image was converted to grayscale (black and white) using a assigned RGB ratio. Subsequently, an ORB (Oriented FAST and Rotated BRIEF) keypoint detector (Rublee et al., 2011; Daradkeh et al., 2021) was used to determine the descriptors of keypoints for each grayscale image, and then calculate the column mean of the descriptors for simple dimensionality reduction ($p = 32$). This process is implemented in JupyterLab 4.0.11 using the OpenCV library and the Python 3.6 programming language.

To evaluate performance, we randomly split the data into equal training and test sets without replacement. Each of the four methods is applied to the training set and assessed on the testing set, repeated 100 times.

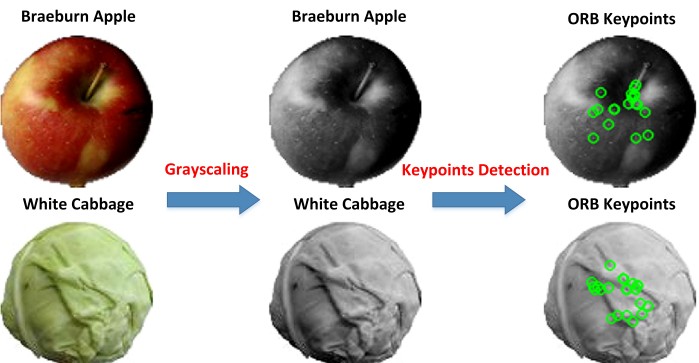

Figure 2: Preprocessing steps for the image dataset.

**Results.** To evaluate algorithm efficacy in the classification task, the following metrics are employed:

$$\text{Specificity} = \frac{\text{TN}}{\text{TN} + \text{FP}}, \quad \text{Sensitivity} = \frac{\text{TP}}{\text{TP} + \text{FN}},$$
$$\text{Precision} = \frac{\text{TP}}{\text{TP} + \text{FP}}, \quad \text{Accuracy} = \frac{\text{TP} + \text{TN}}{\text{TP} + \text{TN} + \text{FP} + \text{FN}},$$

where TP (True Positives) and TN (True Negatives) represent correct classifications for positive (class 1) and negative (class 2) cases, respectively. FP (False Positives) and FN (False Negatives), on the other hand, indicate misclassifications. All these metrics fall within the range of 0 to 1. A higher score suggests that the classification algorithm is performing well, while a lower score indicates subpar performance. An algorithm hitting a perfect score of 1 across the board is essentially considered the gold standard.

Table 3 shows the performance of different methods on the real-world dataset, which gives the four metrics of different methods. From Table 3, we can observe that SSLDA achieving the highest values for Specificity, Precision and Accuracy. Although SSLDA does not achieve the highest Sensitivity value, its performance is comparable to that of the other three methods. This results further confirms the reliability of SSLDA classifier method in high-dimensional scenario.

Table 3: Comparisons of average (standard deviation) classification accuracy of Apple and Cabbage datasets over 100 replications.

| Method | Specificity | Sensitivity | Precision | Accuracy |
|---|---|---|---|---|
| **SSLDA** | **0.9585 (0.0172)** | 0.9271 (0.0217) | **0.8847 (0.0168)** | **0.9505 (0.0112)** |
| LDA-CLIME | 0.9197 (0.0212) | 0.9382 (0.0265) | 0.8001 (0.0431) | 0.9244 (0.0169) |
| CODA | 0.7116 (0.0461) | **0.9668 (0.0216)** | 0.5340 (0.0381) | 0.7761 (0.0336) |
| LS-LDA | 0.9475 (0.0158) | 0.8841 (0.0430) | 0.8524 (0.0363) | 0.9315 (0.0134) |
| AdaLDA | 0.8954 (0.0204) | 0.9418 (0.0276) | 0.7544 (0.0364) | 0.9071 (0.0155) |

## 6 DISCUSSION AND LIMITATION

Reliable classification for high-dimensional data, especially in the case of heavy-tail or untidy data, is typically a tricky for the applied statistician. This paper proposed a robust classification approach that capable of handling heavy-tailed data. The findings from the simulation trials showcase its superior enhanced finite sample prowess and notable computational efficiency both in synthetic data and real data. The primary limitation of this approach is its reliance on a stringent assumption of elliptical symmetric distribution.

## REPRODUCIBILITY STATEMENT

The proposed SSLDA algorithm and models are described in the main text and appendix. All source code and datasets used in our experiments are publicly available on https://github.com/RobustC/SSLDA, with preprocessing steps and parameter settings fully documented. This ensures that the results reported in this paper can be reproduced.

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

# Supplementray Material of "Spatial Sign based Direct Sparse Linear Discriminant Analysis for High Dimensional Data"

## A    LEMMA

**LEMMA A.1.** *Under (C1) and (C2), we have with probability greater than* $1 - O(p^{-1})$,

$$\|p\hat{\mathbf{S}}\boldsymbol{\Lambda}^{-1}\boldsymbol{\delta} - (\tilde{\boldsymbol{\mu}}_1 - \tilde{\boldsymbol{\mu}}_2)\|_\infty \leq \lambda_n \tag{8}$$

*Proof.* By the triangle inequality, we have

$$\|p\hat{\mathbf{S}}\boldsymbol{\Lambda}^{-1}\boldsymbol{\delta} - (\tilde{\boldsymbol{\mu}}_1 - \tilde{\boldsymbol{\mu}}_2)\|_\infty$$
$$\leq \|p\hat{\mathbf{S}}\boldsymbol{\Lambda}^{-1}\boldsymbol{\delta} - p\mathbf{S}\boldsymbol{\Lambda}^{-1}\boldsymbol{\delta}\|_\infty + \|p\mathbf{S}\boldsymbol{\Lambda}^{-1}\boldsymbol{\delta} - \boldsymbol{\delta}\|_\infty + \|\tilde{\boldsymbol{\mu}}_1 - \boldsymbol{\mu}_1\|_\infty + \|\tilde{\boldsymbol{\mu}}_2 - \boldsymbol{\mu}_2\|_\infty$$

According to the proof of Theorem 7.1 in Feng (2024), we have

$$\|\tilde{\boldsymbol{\mu}}_k - \boldsymbol{\mu}_k\|_\infty \leq C\sqrt{\log p/n}, k = 1, 2, \|p\hat{\mathbf{S}} - p\mathbf{S}\|_\infty \leq C\sqrt{\log p/n}$$

for some large enough constant $C > 0$ with probability larger than $1 - O(p^{-1})$. So

$$\|p\hat{\mathbf{S}}\boldsymbol{\Lambda}^{-1}\boldsymbol{\delta} - p\mathbf{S}\boldsymbol{\Lambda}^{-1}\boldsymbol{\delta}\|_\infty \leq \|\boldsymbol{\Lambda}^{-1}\boldsymbol{\delta}\|_1 \|p\hat{\mathbf{S}} - p\mathbf{S}\|_\infty \leq \lambda_n/4.$$

Additionally,

$$\|p\mathbf{S}\boldsymbol{\Lambda}^{-1}\boldsymbol{\delta} - \boldsymbol{\delta}\|_\infty = \|(p\mathbf{S} - \boldsymbol{\Lambda})\boldsymbol{\Lambda}^{-1}\boldsymbol{\delta}\|_\infty \leq \|\boldsymbol{\Lambda}^{-1}\boldsymbol{\delta}\|_1 \|p\mathbf{S} - \boldsymbol{\Lambda}\|_\infty \leq \lambda_n/4$$

by the assumption. $\square$

## B    PROOF OF THEOREMS

### B.1    PROOF OF THEOREM 3.1

*Proof.* By the definition of $\hat{\boldsymbol{\gamma}}$, we have

$$\left|p(\boldsymbol{\Lambda}^{-1}\boldsymbol{\delta})'\hat{\mathbf{S}}\hat{\boldsymbol{\gamma}} - (\boldsymbol{\Lambda}^{-1}\boldsymbol{\delta})'(\tilde{\boldsymbol{\mu}}_1 - \tilde{\boldsymbol{\mu}}_2)\right| \leq \lambda_n \left\|\boldsymbol{\Lambda}^{-1}\boldsymbol{\delta}\right\|_1 + \|\tilde{\boldsymbol{\delta}} - \boldsymbol{\delta}\|_\infty \left\|\boldsymbol{\Lambda}^{-1}\boldsymbol{\delta}\right\|_1$$
$$\leq 2\lambda_n \left\|\boldsymbol{\Lambda}^{-1}\boldsymbol{\delta}\right\|_1. \tag{9}$$

By (8), we have

$$\left|p(\boldsymbol{\Lambda}^{-1}\boldsymbol{\delta})'\hat{\mathbf{S}}\hat{\boldsymbol{\gamma}} - \boldsymbol{\delta}'\hat{\boldsymbol{\gamma}}\right| \leq \lambda_n \|\hat{\boldsymbol{\gamma}}\|_1 + \|\tilde{\boldsymbol{\delta}} - \boldsymbol{\delta}\|_\infty \|\hat{\boldsymbol{\gamma}}\|_1 \leq 2\lambda_n \left\|\boldsymbol{\Lambda}^{-1}\boldsymbol{\delta}\right\|_1, \tag{10}$$

and together with (9) implies

$$|(\hat{\boldsymbol{\gamma}} - \boldsymbol{\Lambda}^{-1}\boldsymbol{\delta})'\boldsymbol{\delta}| \leq 4\lambda_n \left\|\boldsymbol{\Lambda}^{-1}\boldsymbol{\delta}\right\|_1. \tag{11}$$

Then we have

$$|(\tilde{\boldsymbol{\mu}} - \boldsymbol{\mu}_1)'\hat{\boldsymbol{\gamma}} + \frac{1}{2}\boldsymbol{\delta}'\boldsymbol{\Lambda}^{-1}\boldsymbol{\delta}| \leq |(\tilde{\boldsymbol{\mu}} - \boldsymbol{\mu})'\hat{\boldsymbol{\gamma}}| + \frac{1}{2}|\boldsymbol{\delta}'\hat{\boldsymbol{\gamma}} - \boldsymbol{\delta}'\boldsymbol{\Lambda}^{-1}\boldsymbol{\delta}|$$
$$\leq |(\tilde{\boldsymbol{\mu}} - \boldsymbol{\mu})'\hat{\boldsymbol{\gamma}}| + 2\lambda_n \left\|\boldsymbol{\Lambda}^{-1}\boldsymbol{\delta}\right\|_1$$
$$\leq C\sqrt{\frac{\log p}{n}} \left\|\boldsymbol{\Lambda}^{-1}\boldsymbol{\delta}\right\|_1 + 2\lambda_n \left\|\boldsymbol{\Lambda}^{-1}\boldsymbol{\delta}\right\|_1. \tag{12}$$

Similarly, we have

$$|(\tilde{\boldsymbol{\mu}} - \boldsymbol{\mu}_2)'\hat{\boldsymbol{\gamma}} - \frac{1}{2}\boldsymbol{\delta}'\boldsymbol{\Lambda}^{-1}\boldsymbol{\delta}| \leq C\sqrt{\frac{\log p}{n}} \left\|\boldsymbol{\Lambda}^{-1}\boldsymbol{\delta}\right\|_1 + 2\lambda_n \left\|\boldsymbol{\Lambda}^{-1}\boldsymbol{\delta}\right\|_1. \tag{13}$$

Next, considering the denominator in $R_n$. We have

$$\|\boldsymbol{\Sigma}\hat{\boldsymbol{\gamma}} - \boldsymbol{\delta}\|_\infty \leq \|\boldsymbol{\Sigma}\hat{\boldsymbol{\gamma}} - p\hat{\mathbf{S}}\hat{\boldsymbol{\gamma}}\|_\infty + 2\lambda_n \leq C \left\|\boldsymbol{\Lambda}^{-1}\boldsymbol{\delta}\right\|_1 \sqrt{\frac{\log p}{n}} + 2\lambda_n.$$

Thus we have

$$|\hat{\boldsymbol{\gamma}}'\boldsymbol{\Sigma}\hat{\boldsymbol{\gamma}} - \hat{\boldsymbol{\gamma}}'\boldsymbol{\delta}| \le C \left\|\boldsymbol{\Lambda}^{-1}\boldsymbol{\delta}\right\|_1^2 \sqrt{\frac{\log p}{n}} + 2\lambda_n \left\|\boldsymbol{\Lambda}^{-1}\boldsymbol{\delta}\right\|_1.$$

According to (11), we have

$$|\hat{\boldsymbol{\gamma}}'\boldsymbol{\Sigma}\hat{\boldsymbol{\gamma}} - \boldsymbol{\delta}'\boldsymbol{\Lambda}^{-1}\boldsymbol{\delta}| \le C \left\|\boldsymbol{\Lambda}^{-1}\boldsymbol{\delta}\right\|_1^2 \sqrt{\frac{\log p}{n}} + 6\lambda_n \left\|\boldsymbol{\Lambda}^{-1}\boldsymbol{\delta}\right\|_1. \tag{14}$$

Suppose $\boldsymbol{\delta}'\boldsymbol{\Lambda}^{-1}\boldsymbol{\delta} \ge M$ for some $M > 0$. By (6), (12) and (14), we have

$$\left|\frac{(\tilde{\boldsymbol{\mu}} - \boldsymbol{\mu}_1)'\hat{\boldsymbol{\gamma}}}{\sqrt{\hat{\boldsymbol{\gamma}}'\boldsymbol{\Sigma}\hat{\boldsymbol{\gamma}}}}\right| \ge C \left|\frac{\boldsymbol{\delta}'\boldsymbol{\Lambda}^{-1}\boldsymbol{\delta}}{\sqrt{\hat{\boldsymbol{\gamma}}'\boldsymbol{\Sigma}\hat{\boldsymbol{\gamma}}}}\right| \ge C((\boldsymbol{\delta}'\boldsymbol{\Lambda}^{-1}\boldsymbol{\delta})^{-1} + o(1))^{-1/2} \ge CM^{1/2},$$

this implies that

$$|R_n - R| \le \exp(-CM). \tag{15}$$

Suppose $\boldsymbol{\delta}'\boldsymbol{\Lambda}^{-1}\boldsymbol{\delta} \le M$, by (6), and (14), yields

$$\left|\frac{\hat{\boldsymbol{\gamma}}'\boldsymbol{\Sigma}\hat{\boldsymbol{\gamma}}}{\boldsymbol{\delta}'\boldsymbol{\Lambda}^{-1}\boldsymbol{\delta}} - 1\right| = o(1). \tag{16}$$

And together with (12), we have

$$\left|\frac{(\tilde{\boldsymbol{\mu}} - \boldsymbol{\mu}_1)'\hat{\boldsymbol{\gamma}}}{\sqrt{\hat{\boldsymbol{\gamma}}'\boldsymbol{\Sigma}\hat{\boldsymbol{\gamma}}}} + \frac{(1/2)\boldsymbol{\delta}'\boldsymbol{\Lambda}^{-1}\boldsymbol{\delta}}{\sqrt{\hat{\boldsymbol{\gamma}}'\boldsymbol{\Sigma}\hat{\boldsymbol{\gamma}}}}\right| \le C\frac{|\boldsymbol{\Lambda}^{-1}\boldsymbol{\delta}|}{(\boldsymbol{\delta}'\boldsymbol{\Lambda}^{-1}\boldsymbol{\delta})^{1/2}}\lambda_n. \tag{17}$$

By (14), we have

$$\left|\frac{1}{\sqrt{\hat{\boldsymbol{\gamma}}'\boldsymbol{\Sigma}\hat{\boldsymbol{\gamma}}}} - \frac{1}{\sqrt{\boldsymbol{\delta}'\boldsymbol{\Lambda}^{-1}\boldsymbol{\delta}}}\right| \le \frac{C\|\boldsymbol{\Lambda}^{-1}\boldsymbol{\delta}\|_1^2\sqrt{\log p/n} + 6\|\boldsymbol{\Lambda}^{-1}\boldsymbol{\delta}\|_1\lambda_n}{\sqrt{\hat{\boldsymbol{\gamma}}'\boldsymbol{\Sigma}\hat{\boldsymbol{\gamma}}}\sqrt{\boldsymbol{\delta}'\boldsymbol{\Lambda}^{-1}\boldsymbol{\delta}}\left(\sqrt{\hat{\boldsymbol{\gamma}}'\boldsymbol{\Sigma}\hat{\boldsymbol{\gamma}}} + \sqrt{\boldsymbol{\delta}'\boldsymbol{\Lambda}^{-1}\boldsymbol{\delta}}\right)}$$

$$\le C(\boldsymbol{\delta}'\boldsymbol{\Lambda}^{-1}\boldsymbol{\delta})^{-3/2}\left(\|\boldsymbol{\Lambda}^{-1}\boldsymbol{\delta}\|_1^2\sqrt{\frac{\log p}{n}} + \|\boldsymbol{\Lambda}^{-1}\boldsymbol{\delta}\|_1\lambda_n\right). \tag{18}$$

and

$$\left|\frac{(1/2)\boldsymbol{\delta}'\boldsymbol{\Lambda}^{-1}\boldsymbol{\delta}}{\sqrt{\hat{\boldsymbol{\gamma}}'\boldsymbol{\Sigma}\hat{\boldsymbol{\gamma}}}} - \frac{1}{2}(\boldsymbol{\delta}'\boldsymbol{\Lambda}^{-1}\boldsymbol{\delta})^{1/2}\right| \le C\frac{\|\boldsymbol{\Lambda}^{-1}\boldsymbol{\delta}\|_1^2}{(\boldsymbol{\delta}'\boldsymbol{\Lambda}^{-1}\boldsymbol{\delta})^{1/2}}\sqrt{\frac{\log p}{n}} + C\frac{\|\boldsymbol{\Lambda}^{-1}\boldsymbol{\delta}\|_1}{(\boldsymbol{\delta}'\boldsymbol{\Lambda}^{-1}\boldsymbol{\delta})^{1/2}}\lambda_n =: r_n. \tag{19}$$

By condition 1, (17) and (19),

$$R_n = R \times (1 + O(1)r_n(\boldsymbol{\delta}'\boldsymbol{\Lambda}^{-1}\boldsymbol{\delta})^{1/2}\exp(O(1)(\boldsymbol{\delta}'\boldsymbol{\Lambda}^{-1}\boldsymbol{\delta})^{1/2}r_n)). \tag{20}$$

By the assumption $\boldsymbol{\delta}'\boldsymbol{\Lambda}^{-1}\boldsymbol{\delta} \le M$ and the condition (6), we have $(\|\boldsymbol{\Omega}\boldsymbol{\delta}\|_1 + \|\boldsymbol{\Omega}\boldsymbol{\delta}\|_1^2)\sqrt{\log p/n} = o(1)$, thus $R_n = (1 + o(1))R$, when letting $n, p \to \infty$ first and then $M \to \infty$. The proof is completed. $\square$

## B.2 PROOF OF THEOREM 3.2

*Proof.* According to Lemma 6 in Lu & Feng (2025), we know $\|p\mathbf{S} - \boldsymbol{\Sigma}\|_\infty = O(p^{-1/2})$. Additionally, by Lemma 7 in Lu & Feng (2025), we have $\|p\hat{\mathbf{S}} - p\mathbf{S}\|_\infty = O_p(\sqrt{\log p/n})$. Thus, $\|p\hat{\mathbf{S}} - \boldsymbol{\Sigma}\|_\infty = O_p(\sqrt{\log p/n})$ if $\frac{n}{p\log p} \to 0$. Since

$$\|\boldsymbol{\Sigma}(\hat{\boldsymbol{\gamma}} - \boldsymbol{\Lambda}^{-1}\boldsymbol{\delta})\|_\infty \le \|p\hat{\mathbf{S}}(\hat{\boldsymbol{\gamma}} - \boldsymbol{\Lambda}^{-1}\boldsymbol{\delta})\|_\infty + \|(p\hat{\mathbf{S}} - \boldsymbol{\Sigma})(\hat{\boldsymbol{\gamma}} - \boldsymbol{\Lambda}^{-1}\boldsymbol{\delta})\|_\infty$$

$$\le 2\lambda_n + C\|\hat{\boldsymbol{\gamma}} - \boldsymbol{\Lambda}^{-1}\boldsymbol{\delta}\|_1\sqrt{\frac{\log p}{n}}$$

$$\le 2\lambda_n + C\|\boldsymbol{\Lambda}^{-1}\boldsymbol{\delta}\|_0\sqrt{\frac{\log p}{n}}\|\hat{\boldsymbol{\gamma}} - \boldsymbol{\Lambda}^{-1}\boldsymbol{\delta}\|_\infty$$

$$\le 2\lambda_n + C\|\boldsymbol{\Lambda}^{-1}\|_{L_1}\|\boldsymbol{\Lambda}^{-1}\boldsymbol{\delta}\|_0\sqrt{\frac{\log p}{n}}\|\boldsymbol{\Sigma}(\hat{\boldsymbol{\gamma}} - \boldsymbol{\Lambda}^{-1}\boldsymbol{\delta})\|_\infty, \tag{21}$$

together with $\|\boldsymbol{\Lambda}^{-1}\|_{L_1}\|\boldsymbol{\Lambda}^{-1}\boldsymbol{\delta}\|_0\sqrt{\frac{\log p}{n}} = o(1)$ implies that $\|\boldsymbol{\Sigma}(\hat{\boldsymbol{\gamma}} - \boldsymbol{\gamma})\|_\infty \le C\lambda_n$, we have

$$|\hat{\boldsymbol{\gamma}}'\boldsymbol{\Sigma}\hat{\boldsymbol{\gamma}} - \hat{\boldsymbol{\gamma}}'\boldsymbol{\Sigma}\boldsymbol{\Lambda}^{-1}\boldsymbol{\delta}| \le C\|\boldsymbol{\Lambda}^{-1}\boldsymbol{\delta}\|_1\lambda_n,$$

and

$$|\hat{\boldsymbol{\gamma}}'\boldsymbol{\Sigma}\boldsymbol{\Lambda}^{-1}\boldsymbol{\delta} - \boldsymbol{\delta}'\boldsymbol{\Lambda}^{-1}\boldsymbol{\delta}| \le C\|\boldsymbol{\Lambda}^{-1}\boldsymbol{\delta}\|_1\lambda_n.$$

The remaining steps refer to the proof of (20), and the proof is completed. $\square$

## C  ADDITIONAL COMPARING METHODS AND RESULTS

We additionally conducted supplementary experiments to provide a more comprehensive evaluation of our proposed method. In addition to the LDA variants and similarity-based methods presented in the main simulation study, we compared our approach with several widely used general-purpose machine learning classifiers, including Random Forest (RF, Breiman (2001)), Extreme Gradient Boosting (XGBoost, Chen et al. (2015)), Support Vector Machine (SVM, Pisner & Schnyer (2020)), Neural Network (NN, Ripley (2007)), K-Nearest Neighbors (KNN, Dasarathy (1991)), LightGBM (LGBM, Ke et al. (2017)), and Logistic Regression (LG, Hosmer Jr et al. (2013)).

The results, summarized in Table 4 (Model 1) and Table 5 (Model 2), show that our method consistently achieves lower classification error rates than these general-purpose classifiers across a range of distributions and model settings. The advantage is especially pronounced in high-dimensional scenarios, highlighting the robustness and adaptability of our SSLDA approach.

We also assessed computational efficiency (Table 6). Our method is competitive in terms of running time, often comparable to or faster than standard classifiers and other LDA-based or similarity-based methods. These supplementary experiments demonstrate that our approach achieves a favorable balance between accuracy and efficiency, reinforcing its practical value for high-dimensional classification tasks.

Table 4: The average classification error and the standard error (in brackets) for the test samples in percentage for Model 1 over 100 Monte Carlo replications. (%)

| Distribution | $p$ | RF | XGBoost | SVM | NN | KNN | LGBM | LG |
|---|---|---|---|---|---|---|---|---|
| Normal | 100 | 7.60(0.28) | 10.85(0.28) | 2.66(0.14) | 7.93(0.28) | 10.69(0.29) | 10.76(0.29) | 8.15(0.26) |
| | 200 | 7.56(0.27) | 9.23(0.28) | 5.15(0.22) | 11.25(0.32) | 14.58(0.36) | 8.58(0.26) | 16.88(0.43) |
| | 300 | 9.75(0.36) | 11.19(0.34) | 5.47(0.23) | 14.41(0.42) | 17.61(0.35) | 10.71(0.34) | 45.94(0.61) |
| | 500 | 12.38(0.43) | 12.01(0.30) | 12.35(0.36) | 21.66(0.35) | 25.33(0.43) | 11.18(0.27) | 45.73(0.60) |
| | 800 | 11.27(0.34) | 12.60(0.34) | 14.82(0.41) | 24.79(0.41) | 32.40(0.36) | 11.63(0.30) | 46.32(0.57) |
| $t_2$ | 100 | 18.35(0.34) | 18.95(0.34) | 13.46(0.29) | 17.06(0.31) | 21.60(0.39) | 17.88(0.32) | 17.45(0.34) |
| | 200 | 18.02(0.36) | 19.10(0.36) | 12.87(0.32) | 18.18(0.29) | 28.19(0.49) | 18.35(0.37) | 25.70(0.40) |
| | 300 | 17.85(0.36) | 17.38(0.33) | 14.20(0.30) | 20.44(0.37) | 28.99(0.46) | 17.15(0.32) | 47.70(0.51) |
| | 500 | 18.25(0.35) | 18.80(0.33) | 28.45(0.70) | 27.02(0.42) | 42.30(0.48) | 18.46(0.32) | 45.70(0.58) |
| | 800 | 20.35(0.40) | 20.26(0.38) | 41.25(0.63) | 29.78(0.42) | 40.12(0.46) | 19.50(0.32) | 47.48(0.57) |
| $MN_{N,\kappa,10}$ | 100 | 14.99(0.38) | 17.75(0.33) | 12.48(0.32) | 11.43(0.24) | 17.15(0.33) | 16.96(0.34) | 16.81(0.37) |
| | 200 | 15.45(0.37) | 14.95(0.02) | 29.38(0.53) | 10.04(0.26) | 20.39(0.31) | 14.87(0.32) | 25.24(0.43) |
| | 300 | 18.30(0.34) | 18.18(0.34) | 42.35(0.57) | 14.93(0.35) | 26.58(0.36) | 17.28(0.33) | 46.41(0.49) |
| | 500 | 17.00(0.34) | 21.89(0.34) | 48.89(0.41) | 16.87(0.39) | 32.74(0.40) | 19.99(0.35) | 46.40(0.56) |
| | 800 | 17.72(0.45) | 17.99(0.35) | 49.04(0.51) | 16.56(0.36) | 30.47(0.39) | 17.25(0.35) | 47.25(0.56) |
| Cauchy | 100 | 23.50(0.37) | 23.73(0.30) | 49.92(0.25) | 20.25(0.31) | 21.87(0.36) | 23.35(0.35) | 26.30(0.36) |
| | 200 | 23.90(0.38) | 26.12(0.38) | 48.62(0.35) | 23.95(0.35) | 29.70(0.41) | 25.07(0.37) | 26.21(0.40) |
| | 300 | 22.85(0.32) | 23.20(0.37) | 50.35(0.25) | 23.70(0.39) | 32.66(0.46) | 22.73(0.35) | 46.85(0.54) |
| | 500 | 26.40(0.44) | 25.92(0.41) | 48.35(0.25) | 28.53(0.39) | 39.36(0.39) | 25.30(0.39) | 46.95(0.56) |
| | 800 | 25.84(0.42) | 23.87(0.35) | 49.35(0.17) | 31.47(0.39) | 44.43(0.40) | 23.21(0.34) | 47.16(0.46) |

Table 5: The average classification error and the standard error (in brackets) for the test samples in percentage for Model 2 over 100 Monte Carlo replications, excluding AdaLDA. (%)

| Distribution | $p$ | RF | XGBoost | SVM | NN | KNN | LGBM | LG |
|---|---|---|---|---|---|---|---|---|
| Normal | 100 | 23.65(0.33) | 23.04(0.32) | 23.43(0.33) | 25.56(0.41) | 33.71(0.37) | 22.25(0.33) | 31.44(0.39) |
| | 200 | 20.93(0.31) | 21.11(0.34) | 23.15(0.33) | 26.80(0.38) | 33.65(0.37) | 20.83(0.30) | 35.40(0.44) |
| | 300 | 26.38(0.34) | 26.34(0.34) | 29.45(0.39) | 30.84(0.39) | 41.99(0.35) | 25.90(0.36) | 48.91(0.46) |
| | 500 | 26.23(0.30) | 25.67(0.35) | 30.78(0.37) | 36.17(0.45) | 39.84(0.38) | 25.35(0.35) | 47.85(0.48) |
| | 800 | 27.81(0.31) | 30.37(0.35) | 34.91(0.41) | 39.47(0.37) | 44.57(0.42) | 30.10(0.32) | 49.39(0.46) |
| $t_2$ | 100 | 29.78(0.42) | 31.57(0.35) | 31.09(0.35) | 35.08(0.41) | 39.54(0.43) | 30.78(0.36) | 31.10(0.39) |
| | 200 | 27.50(0.40) | 29.79(0.41) | 28.99(0.41) | 32.14(0.40) | 36.75(0.36) | 29.68(0.41) | 37.66(0.45) |
| | 300 | 29.78(0.42) | 31.57(0.35) | 31.09(0.35) | 35.08(0.41) | 39.54(0.43) | 30.78(0.36) | 48.48(0.45) |
| | 500 | 33.25(0.39) | 33.03(0.40) | 38.45(0.44) | 39.38(0.43) | 43.96(0.48) | 31.93(0.37) | 48.49(0.46) |
| | 800 | 27.28(0.35) | 26.53(0.32) | 35.86(0.46) | 39.29(0.41) | 43.32(0.38) | 25.30(0.33) | 48.10(0.54) |
| $MN_{N,\kappa,10}$ | 100 | 29.21(0.40) | 29.21(0.35) | 30.68(0.35) | 29.18(0.37) | 38.06(0.42) | 27.84(0.38) | 37.85 (0.41) |
| | 200 | 28.20(0.40) | 27.30(0.33) | 29.00(0.42) | 27.35(0.39) | 35.74(0.37) | 27.05(0.33) | 43.04(0.42) |
| | 300 | 30.17(0.36) | 32.75(0.38) | 40.86(0.65) | 33.90(0.39) | 41.19(0.40) | 32.19(0.36) | 49.36(0.47) |
| | 500 | 30.67(0.33) | 31.16(0.35) | 44.31(0.58) | 34.86(0.40) | 41.09(0.42) | 30.71(0.37) | 48.73(0.49) |
| | 800 | 28.69(0.33) | 27.55(0.37) | 50.18(0.65) | 36.92(0.39) | 40.50(0.40) | 26.49(0.32) | 49.10(0.41) |
| Cauchy | 100 | 33.60(0.39) | 32.16(0.37) | 48.56(0.47) | 32.87(0.39) | 33.30(0.39) | 32.45(0.42) | 40.86(0.41) |
| | 200 | 34.97(0.42) | 35.37(0.40) | 48.68(0.22) | 37.35(0.36) | 37.93(0.41) | 33.80(0.38) | 42.11(0.41) |
| | 300 | 30.44(0.38) | 32.52(0.39) | 49.40(0.37) | 35.93(0.40) | 40.01(0.45) | 32.02(0.34) | 49.92(0.52) |
| | 500 | 31.15(0.40) | 31.17(0.35) | 50.24(0.21) | 36.95(0.43) | 39.83(0.38) | 30.80(0.35) | 49.28(0.41) |
| | 800 | 29.23(0.37) | 30.04(0.33) | 46.49(0.28) | 38.75(0.45) | 41.72(0.40) | 29.64(0.37) | 47.65(0.46) |

Table 6: Average running time (seconds) of SSLDA and competing methods under the varying model based on 100 replicates. Results are reported for Model 1, with Model 2 values in parentheses.

| Distribution | Method | $p = 100$ | $p = 200$ | $p = 300$ | $p = 500$ | $p = 800$ |
|---|---|---|---|---|---|---|
| Normal | SSLDA | 0.96 (0.06) | 2.45 (0.12) | 6.21 (0.41) | 15.77 (4.27) | 30.49 (36.76) |
| | LDA-CLIME | 0.03 (0.84) | 0.06 (2.13) | 0.43 (5.77) | 16.78 (15.33) | 29.18 (59.34) |
| | CODA | 10.46 (2.89) | 38.31(10.40) | 85.24(23.91) | 39.37 (40.34) | 91.30(140.64) |
| | LS-LDA | 0.85(0.98) | 2.74(2.80) | 6.71(7.10) | 10.49 (13.33) | 18.71 (23.73) |
| | AdaLDA | 0.16(0.14) | 2.81(0.94) | 15.76 (5.94) | 245.64(64.16) | $1.06\times10^3$(392.55) |
| $t_2$ | SSLDA | 1.65 (0.12) | 2.68 (0.08) | 6.83 (0.30) | 17.44 (2.41) | 26.29 (32.22) |
| | LDA-CLIME | 0.38 (0.80) | 0.21 (2.03) | 0.66 (6.04) | 3.37 (15.28) | 35.50(59.31) |
| | CODA | 10.23(2.98) | 38.28(10.48) | 86.07 (24.16) | 39.51 (39.58) | 95.31 (139.22) |
| | LS-LDA | 0.95(1.11) | 3.16(5.21) | 7.46(10.14) | 11.15 (13.12) | 19.35 (23.53) |
| | AdaLDA | 0.15(0.14) | 1.25(0.89) | 17.07(5.59) | 98.75(87.26) | 589.49(447.61) |
| $MN_{N,\kappa,10}$ | SSLDA | 0.07 (0.03) | 0.08 (0.26) | 0.41 (0.65) | 3.07 (4.28) | 20.32 (27.07) |
| | LDA-CLIME | 1.52 (1.27) | 4.21 (3.75) | 12.34 (11.33) | 31.10(27.30) | 75.18(80.49) |
| | CODA | 2.50(3.47) | 8.14(12.01) | 21.79 (21.14) | 52.87 (49.96) | 110.63 (111.49) |
| | LS-LDA | 1.13(1.80) | 3.45 (5.75) | 9.59 (4.97) | 19.77 (6.64) | 45.94(45.58) |
| | AdaLDA | 0.14(0.15) | 1.70(0.99) | 9.76(7.33) | 55.43(49.69) | 305.29(117.55) |
| Cauchy | SSLDA | 0.03 (0.06) | 0.08(0.11) | 0.69(0.33) | 4.31 (1.89) | 22.64 (20.35) |
| | LDA-CLIME | 0.98 (0.83) | 2.43(2.21) | 6.71(6.12) | 26.44(21.72) | 123.49(239.66) |
| | CODA | 10.33(3.09) | 38.34(10.39) | 23.88(24.87) | 39.87 (58.36) | 92.09(138.12) |
| | LS-LDA | 3.68(4.90) | 11.05(16.38) | 15.75(25.42) | 11.36 (6.72) | 9.20 (9.23) |
| | AdaLDA | 0.15(0.14) | 0.95(1.04) | 8.88(5.14) | 33.65(31.72) | 461.86(464.73) |

