# OpenReview forum: "Spatial Sign based Direct Sparse Linear Discriminant Analysis for High Dimensional Data"
_ICLR.cc/2026/Conference — ICLR 2026 Conference Withdrawn Submission_

### Official Review · Reviewer_4A8h · 2025-10-25

**Soundness:** 2
**Presentation:** 2
**Contribution:** 2
**Rating:** 4
**Confidence:** 4

**Summary:**

This paper proposes SSLDA (Spatially Sign-based LDA), a method that constructs discriminant directions using robust estimators of mean and covariance based on spatial signs, which are less sensitive to extreme values than classical moment-based estimators. Unlike traditional LDA, which relies on sample means and covariances (vulnerable to heavy tails), SSLDA uses spatial sign transformations to achieve stability.

**Strengths:**

Unlike traditional LDA, which assumes Gaussian data and is sensitive to outliers, SSLDA leverages spatial sign-based estimators for mean and covariance.

This provides strong theoretical guarantees, ensuring that SSLDA performs well even in high-dimensional settings.

Simulation studies and real-data experiments demonstrate that SSLDA outperforms state-of-the-art robust LDA methods.

**Weaknesses:**

SSLDA relies on the assumption that data follows an elliptical distribution.

The performance of SSLDA depends on the choice of spatial sign scaling parameters.

 It is crucial to explore the model's extension to multi-class scenarios. Additionally, the paper does not explicitly address or discuss the computational complexity associated with SSLDA.

The experimental evaluation on real-world datasets remains limited.

**Questions:**

Some robust LDA such as regularized LDA and L1-norm LDA should be addressed

---

### Official Review · Reviewer_VSmU · 2025-10-28

**Soundness:** 3
**Presentation:** 2
**Contribution:** 2
**Rating:** 4
**Confidence:** 3

**Summary:**

This paper introduces Spatial Sign-based Direct Sparse Linear Discriminant Analysis (SSLDA), a novel classification method designed to address the critical challenge of robust high-dimensional classification under heavy-tailed distributions. The authors identify that classical Linear Discriminant Analysis (LDA) and its high-dimensional sparse variants often fail when data deviates from the Gaussian assumption, as they rely on non-robust sample mean and covariance estimators.

**Strengths:**

This paper presents a robust high-dimensional classifier, Spatial Sign-based Sparse LDA (SSLDA), which directly estimates the optimal discriminant direction under elliptical distributions. The method's core innovation lies in replacing conventional, non-robust estimators with the spatial median and the spatial sign covariance matrix, enabling accurate classification even for heavy-tailed data where standard methods fail. The authors provide strong theoretical guarantees, proving the estimator's consistency and establishing optimal convergence rates for the misclassification error.

**Weaknesses:**

- **1. Limited Discussion on the Elliptical Distribution Assumption:** The paper's entire theoretical framework relies on the assumption that the data follows an elliptical distribution. This is a potential limitation for real-world datasets that may exhibit significant skewness or more complex, non-elliptical dependency structures. The work could be improved by explicitly discussing the robustness of SSLDA to violations of this assumption. A constructive suggestion would be to include a simulation where data is generated from a clearly non-elliptical (e.g., skewed) distribution to empirically explore the method's performance boundaries and better define its applicability.
- **2. Scope of Experimental Validation Could Be Broadened:** Although the experiments are well-designed, they could be more comprehensive to fully demonstrate generalizability. Specifically, the empirical validation relies heavily on synthetic data and a single, specific image classification task. To more convincingly argue for the method's broad utility, the authors could include experiments on a wider range of real-world benchmark datasets from other domains where high-dimensional, heavy-tailed data is common, such as finance (e.g., stock returns), genomics, or text analysis. This would provide stronger evidence of the method's practical impact beyond the presented application.
- **3. Lack of Comparison with Alternative Robust Sparse Methods:** The paper effectively compares SSLDA against several leading sparse LDA methods. However, it does not include comparisons with other classes of robust, high-dimensional classifiers that are not based on the LDA framework, such as robust sparse logistic regression or support vector machines with robust kernels. Including such comparisons would help to position SSLDA more clearly within the broader landscape of robust classification tools and would provide a more complete picture of its relative strengths and weaknesses.

**Questions:**

- **1. On the Robustness Beyond Elliptical Distributions:** The theoretical guarantees of SSLDA are firmly established under the elliptical distribution assumption. Could you please comment on the empirical robustness of SSLDA when this assumption is violated, for instance, with significantly skewed distributions? Have you conducted any preliminary tests on such data? A discussion on the expected behavior or potential modifications to handle non-elliptical data would greatly help users understand the boundaries of the method's applicability.
- **2. On the Generalizability and Practical Impact:** The experimental results on synthetic data and the image classification task are compelling. To further demonstrate the general utility of SSLDA, it would be highly beneficial to see its performance on one or two additional benchmark datasets from domains known for high-dimensional, heavy-tailed data, such as finance (e.g., asset returns) or genomics. This would significantly strengthen the claim of the method's broad practical impact.
- **3. On the Comparison with the Broader Robust Classification Landscape:** The paper provides excellent comparisons against other sparse LDA methods. Could you discuss the rationale for not including comparisons with other paradigms for robust high-dimensional classification, such as $l_1$-regularized robust logistic regression or sparse SVM with robust kernels? A discussion on how you expect SSLDA to perform relative to these alternative approaches, or results from such a comparison, would help to better position your contribution within the entire field of robust classification, not just the LDA family.
- **4. On the Choice of Tuning Parameters:** The method involves a regularization parameter $\lambda_n$. It would be helpful for practitioners if you could provide more detailed guidance on the selection of this parameter in practical scenarios, especially when the underlying distribution is unknown. Did you observe any robust strategies for choosing $\lambda_n$ across different distributional settings in your simulations?

---

### Official Review · Reviewer_pWtR · 2025-10-30

**Soundness:** 3
**Presentation:** 2
**Contribution:** 2
**Rating:** 2
**Confidence:** 4

**Summary:**

This paper focuses on the direct sparse linear discriminant analysis for high-dimensional classification. The proposed SSLDA directly estimates the discriminant direction under the assumption of elliptical distribution, which accelerates the training efficiency without corrupting the final accuracy. Moreover, the spatial sign-based methodology is introduced to handle the heavy-tailed outliers. Theoretical and experimental results demonstrate the superior classification ability of the proposed SSLDA. However, the work seems to be incremental, and the presentation is relatively poor. The detailed comments are summarized in the weakness list. Overall, I think this paper fails to reach the borderline of ICLR.

**Strengths:**

1. This paper introduces the spatial sign-based methodology to the classific LDA algorithm, which can provide a reference for further research.
2. Experiments show the effectiveness of spatial sign-based theory on enhancing LDA.

**Weaknesses:**

1. The proposed SSLDA seems to be a straightforward combination of existing technologies. As discussed in Section 1, the spatial-sign-based methodology is a mature tool for high-dimensional data classification, and it has been integrated into many machine learning algorithms. This paper combines the spatial-sign-based approach with LDA straightforwardly, without making enough novel and significant improvements.
2. In Section 1, the first main contribution states that ‘we establish theoretical results for SSLDA in the sparse scenario’. What exactly does this theoretical result mean? There is a lack of detailed explanation.
3. Section 1 provides a detailed history of LDA with high-dimensional classification, which is somewhat long-winded, and fails to elaborate on the key concepts relevant to SSLDA. What are spatial sign and elliptical distribution? The author should appropriately reduce the review of previous works, and provide a more detailed introduction to SSLDA. It is necessary to clearly and intuitively state why the spatial sign can handle high-dimensional long-tailed distributions.
4. The comparative algorithms are too old. The latest was published in 2019.
5. SSLDA is regarded as a robust classification approach. However, there are no experiments to test the robustness of SSLDA on handling outliers and noisy points.
6. Some equations lack punctuation, such as Eqs. (3) and (5).

**Questions:**

Please see the weakness list. There are no more questions.

---

### Note · Authors · 2025-11-17

I have read and agree with the venue's withdrawal policy on behalf of myself and my co-authors.